# Multispecies and individual gas molecule detection using Stokes solitons in a graphene over-modal microresonator

Teng Tan [1,2,6], Zhongye Yuan[1,6], Hao Zhang[1,6], Guofeng Yan[2], Siyu Zhou[3], Ning An[1], Bo Peng [3], Giancarlo Soavi [4,5✉], Yunjiang Rao [1,2✉] & Baicheng Yao [1✉]

Soliton frequency combs generate equally-distant frequencies, offering a powerful tool for fast and accurate measurements over broad spectral ranges. The generation of solitons in microresonators can further improve the compactness of comb sources. However the geometry and the material's inertness of pristine microresonators limit their potential in applications such as gas molecule detection. Here, we realize a two-dimensional-material functionalized microcomb sensor by asymmetrically depositing graphene in an over-modal microsphere. By using one single pump, spectrally trapped Stokes solitons belonging to distinct transverse mode families are co-generated in one single device. Such Stokes solitons with locked repetition rate but different offsets produce ultrasensitive beat notes in the electrical domain, offering unique advantages for selective and individual gas molecule detection. Moreover, the stable nature of the solitons enables us to trace the frequency shift of the dual-soliton beat-note with uncertainty <0.2 Hz and to achieve real-time individual gas molecule detection in vacuum, via an optoelectronic heterodyne detection scheme. This combination of atomically thin materials and microcombs shows the potential for compact photonic sensing with high performances and offers insights toward the design of versatile functionalized microcavity photonic devices.

[1] Key Laboratory of Optical Fiber Sensing and Communications (Education Ministry of China), University of Electronic Science and Technology of China, Chengdu 611731, China. [2] Research Centre of Optical Fiber Sensing, Zhejiang Laboratory, Hangzhou 310000, China. [3] State Key Laboratory of Electronic Thin Film and Integrated Devices, University of Electronic Science and Technology of China, Chengdu 611731, China. [4] Institute of Solid State Physics, Friedrich Schiller University Jena, Jena 07743, Germany. [5] Abbe Center of Photonics, Friedrich Schiller University Jena, Jena 07745, Germany. [6]These authors contributed equally: Teng Tan, Zhongye Yuan, Hao Zhang. ✉email: giancarlo.soavi@uni-jena.de; yjrao@uestc.edu.cn; yaobaicheng@uestc.edu.cn

Combining ultra-high sensitivity and selectivity is one of the major goals and challenges for photonic detection schemes. In the case of optical sensors aiming at single molecule detection, advanced schemes based on the measurement of the photothermal shift[1], the evanescent wave amplitude in nanowires[2], the plasmonically enhanced scattering[3,4], fluorescent behaviors[5,6], high-$Q$ microresonators[7,8], the mode splitting at the exceptional point[9,10], and the nonlinear response of two dimensional materials[11] have been proposed. In recent years, dual-comb spectroscopy[12–14], which relies on multi-heterodyne detection of two frequency combs with different pulse repetition rates, has enabled real-time identification of gas species[15–18]. Moreover, cogeneration of soliton combs in one single microresonator can be used to improve the compactness of multi-comb sources[19,20] and bridge the lab-to-fab gap. However, selective detection of individual gas molecules using integrated photonic devices is still a challenge because of the stringent requirements in terms of spectral resolution and ultralow noise. The inert nature of the materials (silica, silicon nitride or metal fluorides) that are typically used for soliton microcomb devices inhibits gas adsorption and sensing applications. In this context, chemical functionalization could significantly expand the capability of microcomb devices for sensing applications[21].

Here we demonstrate that functionalization of an over-modal microresonator with a single layer of graphene allows to realize a microcomb sensor with high chemical selectivity (i.e., the capability to distinguish between different chemical species) and high sensitivity (i.e., the capability to measure individual molecules). Such graphene-microcavity device, which has been used in the past for optoelectronic tuning of frequency comb generation[22–25], is used here for gas sensing. In addition, compared to optoelectronic devices, such approach is all-optical and thus does not require a fine tuning of the graphene's Fermi level. Thanks to the over-modal nature of our microresonator, we successfully generate phase-locked Stokes solitons with different frequency offsets that we subsequently use to generate heterodyne beat notes. By depositing graphene 30º away from the equator of the device we can ensure that only the high-order modes of the microcavity will interact with it. Molecule adsorption on graphene changes its Fermi level[22,26] and thus spectrally modifies the high-order Stokes soliton and the beat notes. This approach offers a unique sensitivity for the detection of gas mixtures. Moreover, taking advantage of an advanced heterodyne detection scheme, we trace the frequency shift of a beat-note with sub Hz uncertainty and detect single molecule dynamics.

## Results

### Fabrication and characterization of the hybrid photonic device.

Figure 1a shows the conceptual design of the graphene based micro dual-comb device (GMDC). A silica microsphere with diameter ≈600 μm and typical intrinsic $Q$ factor ≈3 × 10$^8$ is used for the Kerr and Stokes soliton cogeneration. Thanks to its large mode volume, such a microsphere supports multiple transverse co-oscillating intracavity modes, driven by one single pump laser. This enables different soliton frequencies to be generated simultaneously either in a low-order mode (blue arrow) or in high-order modes (yellow). In this architecture, a mechanically exfoliated graphene flake is deposited on the surface of the microsphere by deterministic dry-transfer[27]. The position of graphene is carefully controlled to be 30º above the equatorial plane, to ensure the overlap mainly with the high-order modes (with wider energy distribution) of the microcavity. On the other hand, the fundamental mode, which distributes tightly along the equator, will not be affected by the presence of graphene and/or gas molecules. Such scheme prevents graphene damage and heating

when the intracavity power is high (up to tens of kilowatts). Figure 1b shows a top-view optical microscopy picture of our device and a scanning electron microscopy image where the atomically smooth surface of the silica microresonator and the 80 × 30 μm$^2$ graphene layer are clearly visible. More details about the transfer method and characterization of the device are available in the Supplementary Notes S3 and S4.

Figure 1c plots the optical spectrum of the co-generated Kerr and Stokes solitons. The Kerr soliton spans from 1500 nm to 1600 nm, while the multiple Stokes solitons are generated in the band from 1650 nm to 1700 nm. Although their central wavelengths are different, the Stokes solitons have the same repetition rate, which is locked on the Kerr soliton. In the zoomed-in panel, we also observe that the excited Stokes solitons have distinct comb envelopes, since they belong to different mode families. This means that one Kerr soliton can trap many Stokes solitons thanks to the over-modal nature of the microresonator. As a consequence, these Stokes solitons can beat with each other, offering a powerful tool for multi-channel sensing in the electrical domain. We also show the calculated and measured soliton evolution in the Supplementary Note S1. Figure 1d maps the frequency-resolved auto-correlation traces of the Kerr soliton and the Stokes solitons based on second-harmonic generation (SHG). First, in the C band (1550 nm), the pulse structure with 10.24 ps interval clearly suggests the existence of a single Kerr soliton. Based on the autocorrelation trace, the measured pulse duration of the Kerr soliton is 350 fs, as expected from the 3 dB spectral range of 0.93 THz. On the other hand, in the U band (1670 nm), we can see that there are multiple pulses in one roundtrip. Since the energy per pulse of the Stokes solitons is limited, their signal to noise ratio is lower.

### Gas molecule detection scheme using Stokes solitons.

Figure 2a shows the schematic diagram of our sensing device. As an example, we consider three Stokes solitons (No. 1, 2, and 3) in three different modes. These have different frequency offsets and they can generate three beat notes with frequencies $\Delta f_{1,2}$, $\Delta f_{2,3}$, and $\Delta f_{1,3}$. Once gas molecules are adsorbed on the graphene flake, the refractive index experienced by the solitons will change[26,28], leading to a change of $\Delta f_{1,2}$, $\Delta f_{2,3}$, and $\Delta f_{1,3}$. Finally, we can measure these shifts of the beating frequencies in the electrical domain to obtain the gas dynamics. We note that the resonance shifts for the three modes can be very different since the solitons are generated in different modes. As a result, the frequency offsets of the solitons are modified separately. The $\Delta f_{1,2}$, $\Delta f_{2,3}$, and $\Delta f_{1,3}$ become $\Delta f'_{1,2}$, $\Delta f'_{2,3}$, and $\Delta f'_{1,3}$. Thus, a thorough analysis of $\Delta f'_{1,2} - \Delta f_{1,2}$, $\Delta f'_{2,3} - \Delta f_{2,3}$, and $\Delta f'_{1,3} - \Delta f_{1,3}$, will provide a precise measurement of the concentrations of at least three different species in a gas mixture.

Figure 2b plots the beating signal of the Stokes solitons, measured by a photodetector. These beat notes can not be due to the Kerr–Stokes interaction, as they do not overlap in frequency (photon energy). In 1 GHz band we observe six soliton to soliton beat notes at frequencies 7.514 MHz, 115.12 MHz, 338.37 MHz, 552.88 MHz, 612.05 MHz and 882.46 MHz. Before leveraging the soliton heterodyne signal to detect different gas species, we select three of these beat notes: $f_A = 7.5$ MHz, $f_B = 115$ MHz, and $f_C = 338$ MHz. All the three beat notes show narrow linewidth <10 Hz. The signal-to-noise ratios (SNRs) of $f_A$, $f_B$ and $f_C$ are defined by the different spectral overlaps and are 55 dB, 35 dB, and 46 dB, respectively. Moreover, Fig. 2c shows the measured single-sideband phase noise (SSB-PN) of this beat notes. This reveals that the phase noise of $f_A$ is <−132 dBc/Hz at 10 kHz, and <−140 dBc/Hz at 1 MHz. For $f_B$ and $f_C$, the phase noise is: SSB-PN($f_B$) <−124 dBc/Hz at 10 kHz, and SSB-PN($f_C$) <−118 dBc/Hz

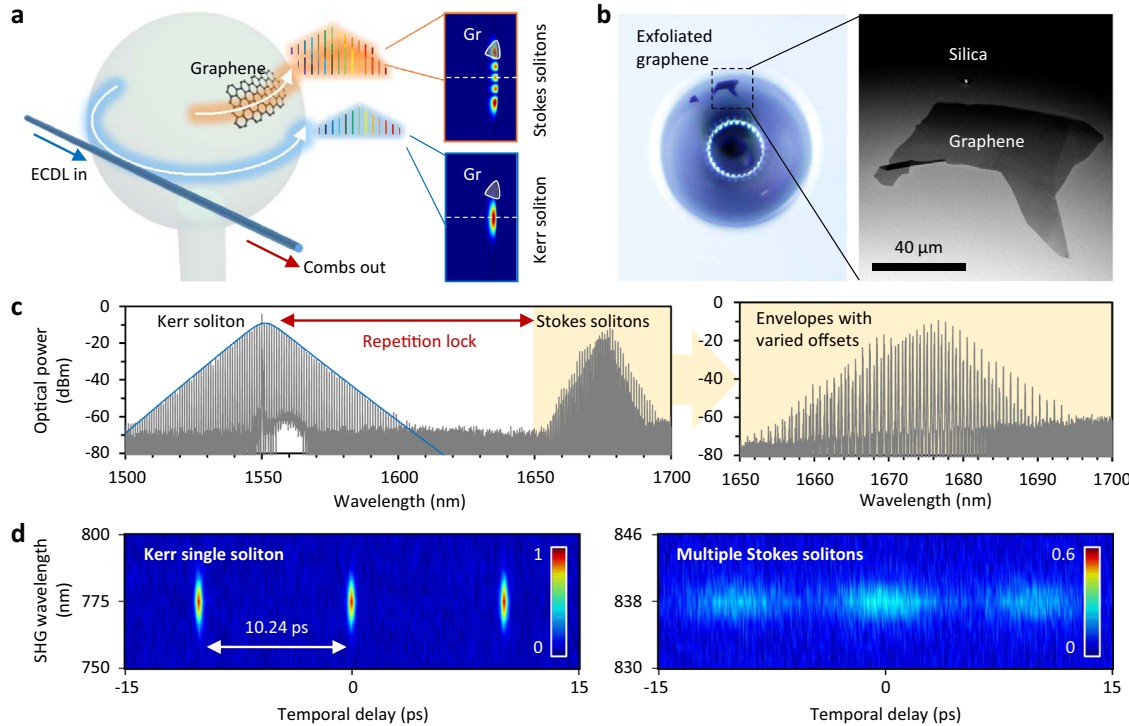

**Fig. 1 Conceptual design of the graphene based over-modal microresonator and the Kerr–Stokes multiple comb cogeneration. a** Schematic diagram of the device. An external cavity diode laser (ECDL) is used as the pump to excite soliton combs, which belong to distinct mode families. Graphene can only influence the comb modes with large spatial distributions. **b** The optical microscopy and SEM pictures show the exfoliated graphene deposited on the microsphere, with area ≈ 80 μm × 30 μm. Scale bar: 40 μm. **c** The measured optical spectrum shows both Kerr soliton and Stokes solitons. The Kerr comb is in the C + L band, and the Stokes comb excited by the Raman gain appear in U band. **d** Frequency-resolved auto-correlation maps of the Kerr soliton (Left panel) and the Stokes solitons (Right panel). The color bar shows the normalized intensity of the autocorrelation traces.

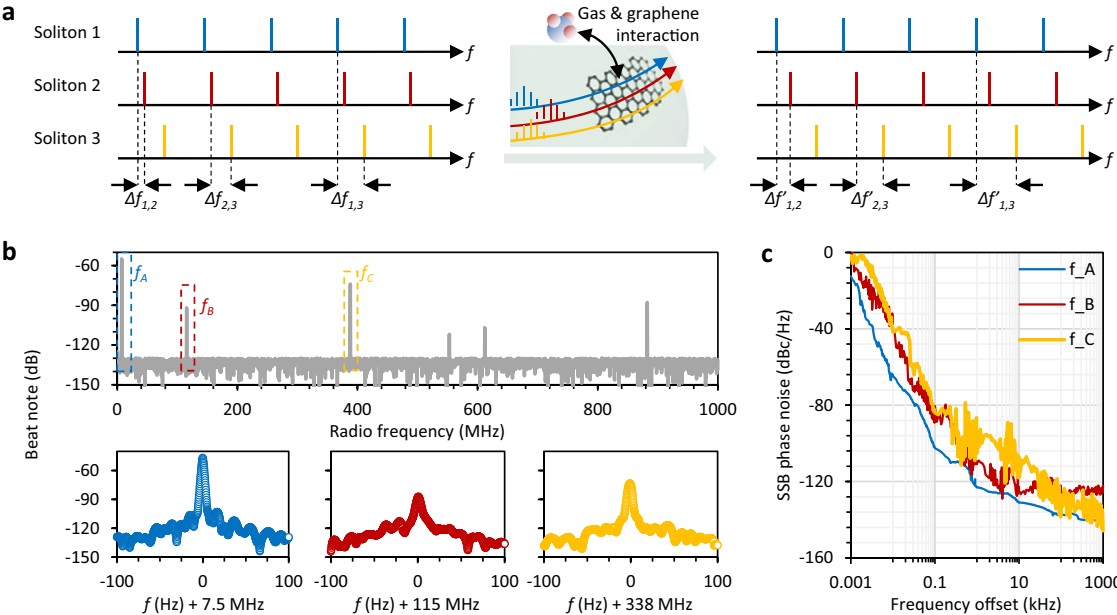

**Fig. 2 Working principle and characterization of the beat notes of the Stokes solitons. a** Schematic diagram of beating frequency generation. **b** Beat note of the Stokes combs in the U band. The Stokes solitons share the same repetition rate but have different frequency offsets. Here we select three beat notes $f_A$ (7.514 MHz), $f_B$ (115.12 MHz), and $f_C$ (338.37 MHz). All of them show narrow linewidth and high signal to noise ratio. **c** Single-sideband phase noise of $f_A$, $f_B$, and $f_C$, down to −140 dBc/Hz.

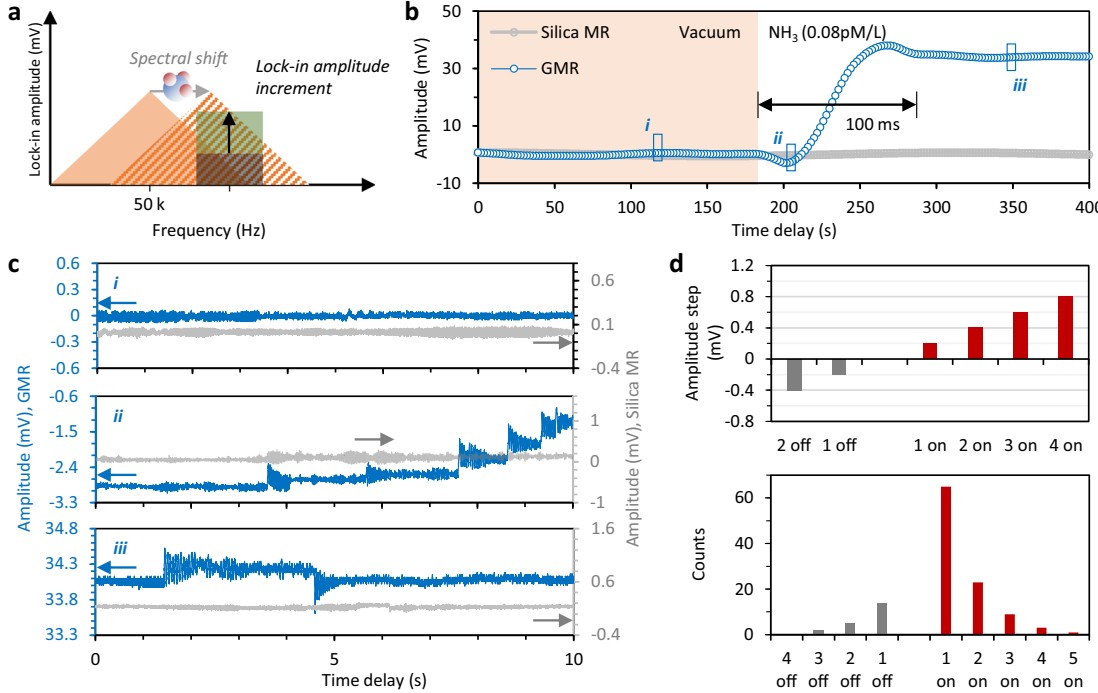

**Fig. 3 Detection of individual molecule dynamics. a** Operating principle of the lock-in amplification for enhanced individual molecule detection. This technique enables us to detect a small frequency shift (sub Hz) as a large (>30 dB) change in intensity. **b** Measured output trace of the lock-in amplifier, the amplitude increment is induced by the $NH_3$ gas absorption on graphene. After injection of the $NH_3$ gas in the chamber, the time window to switch from state i (vacuum) to state iii (dynamic balance) is ≈100 s. MR: microresonator; GMR: graphene microresonator. **c** Zoomed-in panels of **b**. Discrete steps in state ii and iii suggest individual molecule on/off events. In panel **b** and **c**, the gray curve shows the result of a pristine silica microsphere without graphene deposition, for comparison. **d** Top panel: the measured steps are integer multiples of 0.2 mV. Bottom panel: We count the molecular on/off (adsorption/desorption) events. The obtained statistics follow a power law.

at 10 kHz. This result could be further optimized by using active feedback[25,29,30]. In the experiments, we also measured the long-term stability of the beating signals. For a continuous measurement of 2 hours at room temperature the frequency shift is <5 Hz and the intensity fluctuation is <±0.1 dB (Supplementary Note S5). In comparison to conventional sensing schemes based on passive microresonators[8,31], the generation of coherently co-locked Stokes solitons offers ultrahigh frequency resolution and the unique possibility to detect different gas species from a mixture. In this sense, the enhanced performances of our photonic device compensate for its increased complexity.

**Individual gas molecule detection**. First, we use only one beating signal ($f_A$) to realize single molecule detection of $NH_3$. This is done by measuring its spectral shift based on a high-resolution heterodyne lock-in amplification scheme, as shown schematically in Fig. 3a. We use a signal generator to produce a RF line with a stable frequency of 7.464 MHz to beat the $f_A$, thus forming a new 50 kHz frequency ($\Delta f_A$) that falls within the bandwidth of the lock-in amplifier (125 kHz, Stanford Research SR810) used for our experiments. Any shift of the dual-comb beat note will induce a shift of the $\Delta f_A$. We then fix the trigger frequency at 50.005 kHz to monitor the frequency shift induced intensity alteration. The lock-in amplifier only amplifies the amplitude at 50.005 kHz. Since the 50 kHz signal $\Delta f_A$ has a 3 dB linewidth of 10 Hz (defined by the spectral linewidth of the soliton beating) with an electrical intensity 1 mV, the frequency shift dependent amplitude change reaches 0.1 mV/Hz in $sech^2$ approximation. Finally, the lock-in amplification further enhances the sensitivity of our device. Besides, the 10 Hz linewidth also limits the minimum time constant of our lock-in scheme at 0.1 s. To achieve higher speed, one

could use the 'electrical filter + electrical amplifier' scheme to replace the lock-in amplifier, at the cost of more complex operations and higher noise. More details on the mechanism and implementation are shown in the Supplementary Note S6 and S7.

We injected 0.08 pM $NH_3$ into the vacuum chamber (corresponding to concentration of 0.01 pM/L in chamber) and we measured the response of $f_A$ (Fig. 3b). Following interaction between the graphene flake and $NH_3$, the lock-in amplified intensity increases from 0 to 34 mV on a time-scale of approximately 100 s, defined by the gas adsorption process. When we zoom-in the gas response trace of Fig. 3b, we observe clear steps caused by individual molecule adsorption/desorption (Fig. 3c). Before injecting the $NH_3$ gas, the trace is uniformly flat and there is no evidence of any molecular on/off case (state i). Once the $NH_3$ gas is injected in the chamber, we observe that the intensity curve increases in small steps, suggesting that adsorption of individual molecules occurs (state ii). When the interaction between the graphene and the $NH_3$ gas reaches the dynamic balance, the intensity curve becomes flat again, although we can still observe on/off steps due to microscopic molecular thermal motion. In Fig. 3b, c, we also plot the case of a pristine silica microresonator, in which Stokes solitons can also be generated: this shows no sensitivity to $NH_3$ molecules and highlight the crucial role of hybridization of the microcavity with graphene. Figure 3d shows that the height of all the observed steps are multiple integers of 0.2 mV, which is the smallest number corresponding to individual molecule adsorption. This is another strong evidence that individual molecule dynamics can be detected with our device. We also count the molecular on-off cases occurring within 200 s after the injection of the gas in the chamber. Gas adsorption dominates during the state ii, and the

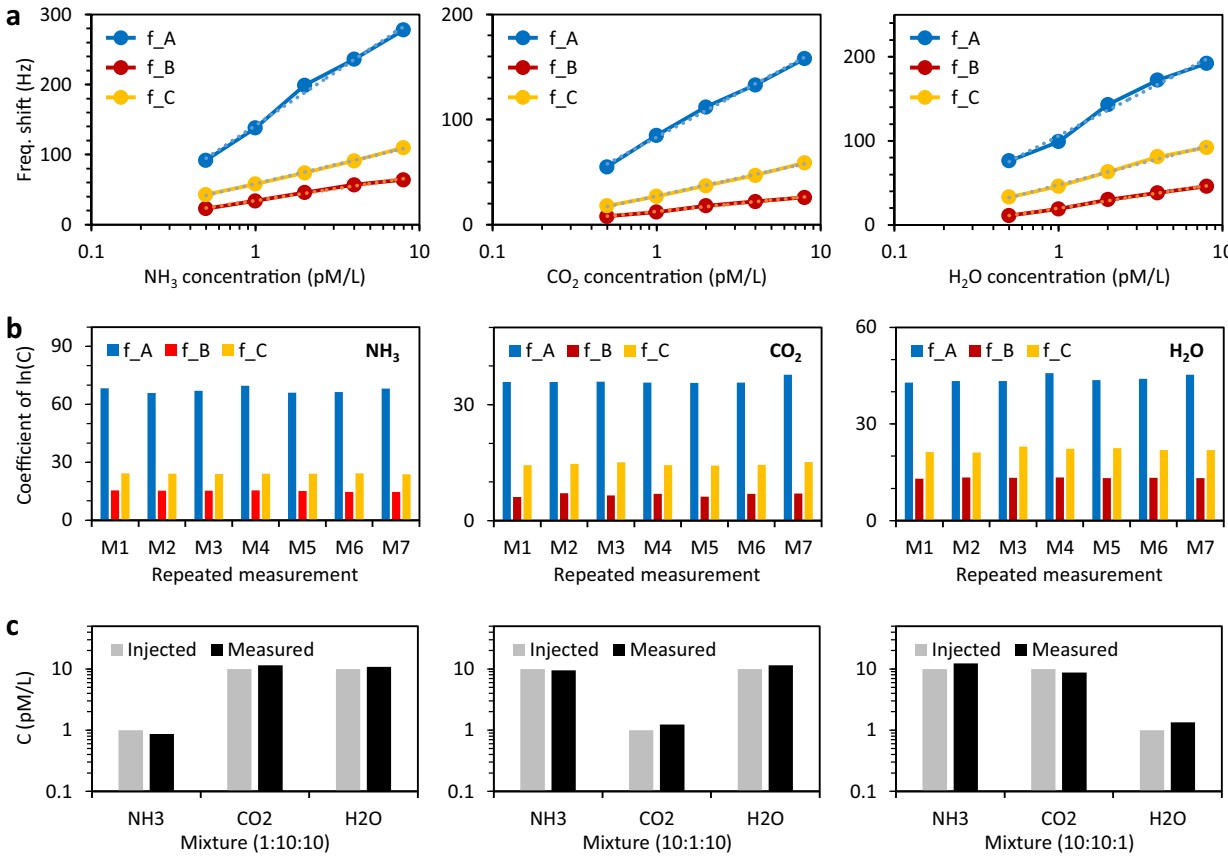

**Fig. 4 Gas identification. a** Gas sensitivity and selectivity: we use the beat notes at $f_A$, $f_B$, and $f_C$ for the detection of $NH_3$, $CO_2$, and $H_2O$ molecules separately. Different intermode beat notes show different 'concentration–spectral shift' correlations (sensitivity coefficients). **b** Repeated measurements show that the sensitivity coefficients of the three beat notes are reliable and reproducible. Generally, this devices shows the highest sensitivity for $NH_3$, determined by the nature of adsorption on graphene. **c** Detection of the three selected gases in a mixture.

| Table 1 Sensitivity Coefficients of 3 selected beat notes. | | | |
| --- | --- | --- | --- |
| | **NH₃** | **CO₂** | **H₂O** |
| $f_A$ | 67.8 | 36.6 | 44.1 |
| $f_B$ | 15.2 | 6.6 | 12.8 |
| $f_C$ | 24.1 | 14.7 | 22.1 |

on-off cases are balanced in the state iii. Moreover, when the graphene microresonator is exposed to $NH_3$, the large steps are rare (such as >2 molecular on/off events), whereas unit steps are dominant. These statistical results obey a power-law distribution, which is also a sign of individual molecule adsorption events.

**Detection of gases in a mixture.** As previously mentioned, a unique advantage of the Stokes soliton sensor is the possibility to simultaneously detect multiple beating frequencies and thus to detect different gas species and achieve high selectivity. Figure 4a shows the performances of our detection scheme when simultaneously measuring $f_A$ (7.514 MHz), $f_B$ (115.12 MHz), and $f_C$ (338.37 MHz) to retrieve the concentrations of three different common gases ($NH_3$, $CO_2$ and $H_2O$) which have been separately injected in the vacuum chamber. The result clearly shows that the different dual-comb beat notes have different responses to the three gases. In addition, the logarithmic correlation of frequency shift *versus* gas concentration tells that the gas sensitivity is lower

when the gas concentration is increasing, as expected from the fact that gas adsorption on graphene can saturate. Figure 4b compares the retrieved sensitivity coefficients (i.e., frequency shift versus gas concentration) $\ln(C)$ obtained from seven different measurements, here $\ln(C)$ is the natural logarithm of the gas concentration. Thanks to the soliton stability, the results demonstrate high fidelity and consistency. These sensitivity coefficients are summarized in Table 1. If we now want to detect simultaneously the three gas samples in a mixture with high selectivity, we simply need to solve the linear equation:

$$\begin{bmatrix} \triangle f_A \\ \triangle f_B \\ \triangle f_C \end{bmatrix} = \begin{bmatrix} 67.8 & 36.6 & 44.1 \\ 15.2 & 6.6 & 12.8 \\ 24.1 & 14.7 & 22.1 \end{bmatrix} \begin{bmatrix} \ln(C_{NH_3}) \\ \ln(C_{CO_2}) \\ \ln(C_{H_2O}) \end{bmatrix} \quad (1)$$

Here $C_{NH3}$, $C_{CO2}$, and $C_{H2O}$ are the gas concentrations in the mixture. Having at our disposal the three beating frequencies of our Stokes solitons, this scheme allows us to quantitatively measure the concentrations of the three gases in the mixture, as showed in Fig. 4c. As a proof of concept, we used the device to detect three different gas mixtures ($NH_3$, $CO_2$, $H_2O$) with component ratio 1:10:10, 10:1:10, and 10:10:1, respectively. Correspondingly, we measured component ratios of 0.87:11.4:10.8, 9.13:1.14:10.9, and 12.2:8.86:1.22. We note that the error could be further reduced via repeated measurements.

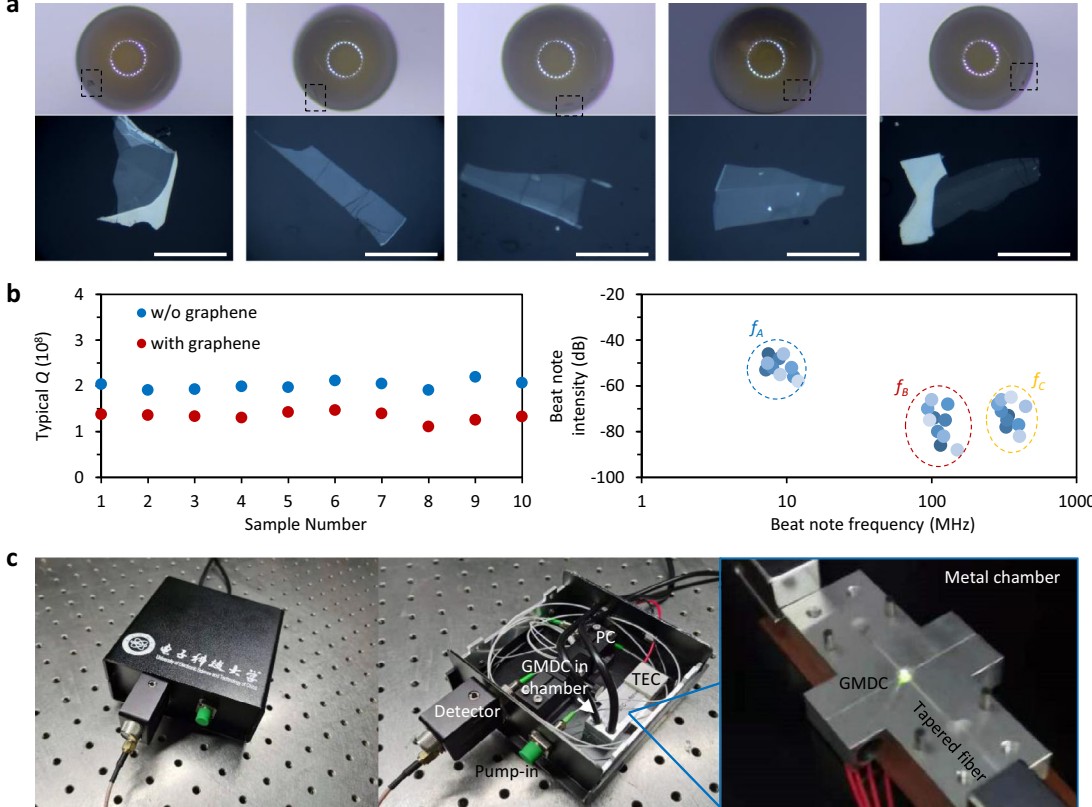

**Fig. 5 Device fabrication reproducibility and compact packaging. a** Top images: fabrication of different graphene based micro dual-comb device (GMDC) samples shows high reproducibility. Bottom pictures: graphene samples deposited on the microspheres. The scale bar is 50 μm. **b** Q factors (left) and measured Stokes soliton–soliton beat notes (right) in ten different devices. **c** Proof of concept of an integrated and miniaturized device, including the GMDC, a Thermo Electric Cooler (TEC), a Polarization Controller (PC), and a photodetector. All the components are linked by optical fibers.

## Discussion

For practical applications, a foreseeable challenge is the degree of reproducibility of real-world devices. In particular, each new device should be calibrated before use due to the strong variations of Stokes solitons and their gas sensing response in different microresonators. To address this issue, we characterized a set of fabrication parameters that could simplify the lab to fab transition. Figure 5a shows the microscope images of five different graphene-silica microsphere samples. Via precise control of the arc-discharge parameters, we always obtained almost identical microsphere devices. Subsequently, graphene samples were deposited on each microsphere, with a size difference in the order of micrometers. Figure 5b compares the Q factor and sensing parameters of ten different devices: all of them have almost identical Q factors, and can provide multiple soliton-soliton beat notes at MHz level. To verify the capability of multispecies gas detection in all devices, we also measured for the ten samples $f_A$, $f_B$ and $f_C$ and obtained highly reproducible results. Finally we demonstrate that, thanks to low-loss fiber links, our graphene-microsphere can be fixed in a miniaturized metal vacuum chamber, and further encapsulated in a ≈10 cm package. A fixed laser diode could be used for soliton excitation, and the optical response of the device is monitored by a single detector (Fig. 5c). This portable device further shows the potential impact of our technology in real-world applications.

To conclude, we demonstrated the cogeneration of multiple Stokes solitons in a graphene functionalized over-modal microresonator, offering an all-optical tool for gas detection with high selectivity and sensitivity. The Stokes soliton combs with ≈1670 nm central wavelength in distinct mode families are co-generated and trapped by the Kerr soliton in the communication band. We can measure the inter-mode beating signals of the Stokes solitons in the in the RF spectral region using only one detector. By placing graphene 30° away from the equator of the microresonator, only the high-order Stokes solitons interact with graphene. Thanks to the stable nature of soliton microcombs, we achieved sub Hz spectral resolution for individual molecule detection. Moreover, the multiple soliton beating in one single device demonstrates unique advantages for the identification of different gases in a mixture. This scheme offers a label-free optical tool to realize quantitative and selective individual gas molecule detection. Such a compact device not only demonstrates a unique potential for chemical sensing, but also paves the way to design microcomb devices for applications ranging from radio signal generators, frequency modulators and spatial rangefinders.

## Methods

**Co-generation of the Kerr and Stokes solitons.** The Kerr comb generation relies on phase-matched four-wave-mixing, while Stokes comb excitation is based on the Raman gain. The photonic energy transfers from the pumping laser and the Kerr comb (C band) to the Stokes lines (U band). Once the laser pump detuning is >5 GHz, the excitation of Stokes combs appear in distinct mode families. (1) By using the finite-element-method, we analyze the mode distributions and refractive indices in the microsphere, and discuss the influence for Kerr–Stokes comb interaction. (2) By using the Lugiato-Lefever-equation with a Raman term, we theoretically analyze the comb formation and trapping and investigate the temporal and spectral evolutions numerically. A detailed discussion is shown in the Supplementary Note S1 and S2.

**Fabrication of the graphene deposited microspheres.** The microsphere resonators are fabricated via electrical arc discharging thermal melting-shaping in a high power fiber fusion splicer. By controlling the arc discharge power, discharge

duration and discharge position, we can control the diameter of the microspheres. Here we use ≈620 μm diameter microspheres because they can support hundreds of transverse mode families. For soliton comb formation, these samples can produce a repetition rate ≈100 GHz. The arc-melting-shaping scheme ensures the surface uniformity and smoothness, enabling ultrahigh $Q$ factor (>10$^8$) for light oscillation. Afterwards, we prepare the high-quality crystalline graphene via PDMS based mechanical exfoliation. Then, by using the dry-transfer technique, we deposit the graphene nano-layer on the surface of the microsphere. In this implementation, we carefully optimize the graphene location, making sure that some modes of the intracavity transmitting light can interact with the graphene, while the graphene is not at the equator of the microresonator. The fabrication steps and characterization of the device are shown in Supplementary Note S3.

**Experimental setups**. We characterize the transmission of our cavities and the $Q$ factor of the resonators by tuning the ECDL wavelength 1550 nm to 1551 nm, fixing the ECDL power at 0.2 mW to avoid any ringdown. The tunable laser is also used as a trigger the time-frequency calibration. The spectral sampling rate can be as low as 1 kHz, which allows to identify resonances in the MHz level. The $Q$ factor corresponds to the carrier frequency times the width of the resonance. For soliton comb formation, we use the power kicking scheme, via detuning the C-band tunable laser diode into the resonances, after erbium based amplification (pumping power 120 mW). Light is launched in and collected by using a tapered fiber with diameter 1 μm. To characterize the pulse duration we use a commercial pulse measurement system (FROGscan, MesaPhotonics). To achieve the SHG threshold, the average comb power is amplified to 1 W. For the optoelectronic sensitivity enhancement, we first use an amplified photodetector (PD, Thorlabs APD 103 C) to extract the dual-comb beat note. Then a wave generator (Tektronix AFG 3100, 250 MHz) is used to produce an electric signal for further down conversion of the comb beat note to 50 kHz. Finally, we use a lock-in amplifier to complete the heterodyne measurement. More details are shown in Supplementary Note S4–S7.

## Data availability

The data that support the plots within this paper and other findings of this study are available from the corresponding author on reasonable request. Source data are provided with this paper.

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

## Acknowledgements

The authors acknowledge support from the National Science Foundation of China (61975025), and the Zhejiang Lab—UESTC collaboration project (202012KFY00562). This work was also supported by the European Union's Horizon 2020 research and innovation program under Grant Agreement GrapheneCore3 881603. G.S. acknowledges the German Research Foundation DFG (CRC 1375 NOA project B5) and the Daimler und Benz foundation for financial support.

## Author contributions

B.Y. and Y.R. led this work. T.T., Z.Y., G.Y. and B.Y. performed the optical measurements. Z.Y. and T.T. fabricated the silica microspheres. Z.Y., S.Z., H.Z. and B.P. performed the graphene deposition and characterization. G.S. and B.Y. performed the theoretical analysis of the graphene optoelectronics. B.Y., T.T. and Z.Y. contributed the principal investigations of the Kerr–Raman soliton generation. N.A., Z.Y., G.Y. and T.T. performed the ultrafast identification and heterodyne measurement. Z.Y., H.Z. and T.T. performed the gas sensing measurements. All the authors discussed and analyzed the results. B.Y., G.S. and Y.R. prepared the paper.

## Funding

## Competing interests

The authors declare no competing interests.
