## [Peer review file · Nature Communications]

Editorial Note: This manuscript has been previously reviewed at another journal that is not operating a transparent peer review scheme. This document only contains reviewer comments and rebuttal letters for versions considered at Nature Communications

REVIEWER COMMENTS

Reviewer #1 (Remarks to the Author):

The authors successfully addressed my previous comments, and the manuscript looks much better. In particular, additional experiment to detect multiple gas species showed the potential of their work. In my opinion, the current manuscript is OK to be published in Nature Communications.

Reviewer #2 (Remarks to the Author):

The manuscript has been substantially revised and additional data makes the manuscript stronger. The new title describes the work better and the confusing framing as "dual-comb spectroscopy" has been revised. In my understanding, the central element is the graphene-enabled sensitivity and the exploitation of differential resonance shifts as a readout mechanism.

The multi-soliton approach that is employed to detect the differential resonance shift is, in my view, unnecessarily complex and distracting (as no multiheterodyne dual-comb spectroscopy is performed). The soliton generation, the characteristics of the Stokes solitons, their spectral overlap and the calibration of the data (not only sensitivity but even the qualitative system response for different gas molecules) depends critically on resonator properties that are hard to precisely control. Predictable performance of the system appears therefore challenging and this should at least be mentioned.

Similar to reviewer 1, I would not be surprised if a similar result could be achieved with much reduced complexity by a continuous-wave laser-based readout, e.g. in combination with a graphene-enabled high-Q crystalline resonator ($Q 10^9-10^{11}$).

Although I have doubts that the presented sensing concept in conjunction with stokes solitons provides a practically useful implementation, the study is nevertheless of interest to the molecular sensing community.

Before I can recommend the manuscript for publication, I would appreciate if, in line with my previous comments, the authors could provide a quantitative discussion on the detection mechanism: A quantitative estimate of the effect one adsorbed molecule has on the soliton-soliton offset (and magnitude of the step signal) is in my view lacking. Such estimate could inspire confidence in the observed step signals and guide future designs. As the authors have through their simulations detailed knowledge on the mode distribution and can precisely control the graphene flake's location, this appears to be a reasonable request.

A minor comment with regard to the "theoretical maximum resolution" of 10^{-5} Hz: If not mistaken, this is based on a speculative 40 dB amplification of the 0.1 mV/Hz signal, without however considering the signal-to-noise ratio. I suggest to explain better why a useful 40 dB amplification is possible (considering the signal-to-noise ratio before and after amplification), or to remove the statement.

Reviewer #3 (Remarks to the Author):

I commend the authors on the technical clarifications in the revised manuscript. One minor fact that the authors could consider is the correctness of Eq. S7 - it should use n_g , the group index instead of n_{eff} . With these revisions, the manuscript is, in my opinion, suitable for Nature Communications.

Response Letter

All the changes are marked in RED in the re-submitted marked version of the manuscript.

Reviewer #1 (Remarks to the Author):

The authors successfully addressed my previous comments, and the manuscript looks much better. In particular, additional experiment to detect multiple gas species showed the potential of their work. In my opinion, the current manuscript is OK to be published in Nature Communications.

Response: We sincerely thank the Reviewer for his/her comments and support to publish in Nature Communications.

Reviewer #2 (Remarks to the Author):

The manuscript has been substantially revised and additional data makes the manuscript stronger. The new title describes the work better and the confusing framing as “dual-comb spectroscopy” has been revised. In my understanding, the central element is the graphene-enabled sensitivity and the exploitation of differential resonance shifts as a readout mechanism.

Response: We thanks the Reviewer for the positive evaluation of the new version of our manuscript and for highlighting the central element of our work. In the following we carefully address all his/her concerns.

1. The multi-soliton approach that is employed to detect the differential resonance shift is, in my view, unnecessarily complex and distracting (as no multiheterodyne dual-comb spectroscopy is performed). Similar to reviewer 1, I would not be surprised if a similar result could be achieved with much reduced complexity by a continuous-wave laser-based readout, e.g. in combination with a graphene-enabled high-Q crystalline resonator (Q 10^9 – 10^{11}).

Response: We thank the Reviewer for the comment. While we agree that the realization of our photonic device is (with current technologies) still challenging, we firmly believe that its implementation provide unique and distinctive advantage in particular in the identification of different gas species. The Referee suggests that similar results could

be obtained using a continuous-wave laser-based readout, in combination with a graphene-enabled high-Q crystalline resonator ($Q \approx 10^9$ – 10^{11}). In this regard, we would like to highlight two aspects that highlight the advantage of our approach:

(1) **Sensitivity.** In the past, detection of single nanoparticles has been demonstrated on high-Q crystalline resonators driven by low power CW lasers (see e.g. 10.1038/nphoton.2009.237). This, can lead to a resonance shift in the MHz level, which is thus suitable for resonator with $Q \approx 10^9$ (corresponding to a resonance width ≈ 200 kHz for the C band). However, in the case of gas molecule detection, the resonance shift is much smaller (sub Hz level) and thus hard to detect/resolve even with cavities of $Q \approx 10^{11}$ (2 kHz width). Moreover, in a CW laser-based readout scheme, the measurement accuracy will be dramatically affected by the drifting uncertainty of the laser itself, typically > 100 kHz, regardless of the cavity's Q factor. In contrast, in our approach the down converted beat note of the Stokes solitons is highly coherent: its linewidth is < 10 Hz and its drifting uncertainty is on the single Hz level. This unique ultrahigh resolution enabled by our approach is the key to achieve single molecule sensitivity. As a further proof, we fabricated and tested the gas sensitivity performances of a pristine high $Q \approx 10^9$ resonator. **Fig. R1** shows no detectable difference in its resonance before and after injection of 10 pM/L of NH_3 .

Fig. R1. Sensing performances of a high Q silica microresonator. Sampling rate: 20 points per MHz.

(2) **Multispecies detection capability.** We agree with the Referee that it is not strictly necessary to generate multiple co-locked Stokes solitons when the goal is to detect a single type of gas molecules. However, if one wants to distinguish different gas species

from a mixture, the co-locked Stokes solitons obtained from different mode families show unique advantages, as we explain in the manuscript.

We have now added the following sentence to the revised version of the manuscript:

“In comparison to conventional sensing schemes based on passive microresonators [10.1038/nmeth.1221; 10.1038/nphoton.2009.237], the generation of coherently co-locked Stokes solitons offers ultrahigh frequency resolution and the unique possibility to detect different gas species from a mixture. In this sense, the enhanced performances of our photonic device compensate for its increased complexity.”

2. The soliton generation, the characteristics of the Stokes solitons, their spectral overlap and the calibration of the data (not only sensitivity but even the qualitative system response for different gas molecules) depends critically on resonator properties that are hard to precisely control. Predictable performance of the system appears therefore challenging and this should at least be mentioned. Although I have doubts that the presented sensing concept in conjunction with stokes solitons provides a practically useful implementation, the study is nevertheless of interest to the molecular sensing community.

Response: We thank the Referee for the comment and for acknowledging that “the study is of interest to the molecular sensing community”. We agree that the performances of our device strongly depend on the characteristic of each specific sample/resonator, leading to possible shortcomings in terms of reproducibility. To address this comment, we now try to provide a larger statistics on the fabrication procedure and its consistency (**Figure 5a-b**). Looking in the direction of a practically useful implementation, we also prepared a prototype of a compact gas sensing device for possible out-of-lab applications (**Figure 5c**).

We thus added the following paragraph and figure to the revised version of the manuscript:

“For practical applications, a foreseeable challenge is the degree of reproducibility of real-world devices. In particular, each new device should be calibrated before use due to the strong variations of Stokes solitons and their gas sensing response in different microresonators. To address this issue, we characterized a set of fabrication parameters

that could simplify the lab to fab transition. **Fig. 5a** shows the microscope images of five different graphene-silica microsphere samples. Via precise control of the arc-discharge parameters, we always obtained almost identical microsphere devices. Subsequently, graphene samples were deposited on each microsphere, with a size difference in the order of micrometers. **Fig. 5b** compares the Q factor and sensing parameters of ten different devices: all of them have almost identical Q factors, and can provide multiple soliton-soliton beat notes at MHz level. To verify the capability of multispecies gas detection in all devices, we also measured for the ten samples f_A , f_B and f_C and obtained highly reproducible results. Finally we demonstrate that, thanks to low-loss fiber links, our graphene-microsphere can be fixed in a miniaturized metal vacuum chamber, and further encapsulated in a ≈ 10 centimeter package. A fixed laser diode could be used for soliton excitation, and the optical response of the device is monitored by a single detector (**Fig. 5c**). This portable device further shows the potential impact of our technology in real-world applications.”

Figure 5 | Device fabrication reproducibility and compact packaging. a, Top images: fabrication of different GMDC samples shows high reproducibility. Bottom pictures: graphene samples deposited on the microspheres. The scale bar is $50 \mu\text{m}$. **b**, Q factors (left) and measured Stokes soliton-soliton beat notes (right) in ten different devices. **c**,

Proof of concept of an integrated and miniaturized device, including the GMDC, a TEC, a PC, and a photodetector. All the components are linked by optical fibers.

3. Before I can recommend the manuscript for publication, I would appreciate if, in line with my previous comments, the authors could provide a quantitative discussion on the detection mechanism: A quantitative estimate of the effect one adsorbed molecule has on the soliton-soliton offset (and magnitude of the step signal) is in my view lacking. Such estimate could inspire confidence in the observed step signals and guide future designs. As the authors have through their simulations detailed knowledge on the mode distribution and can precisely control the graphene flake's location, this appears to be a reasonable request.

Response: We thank the Referee for the comment. We have now added the requested information in the revised version of the supplementary note S6.

“In this case, we can quantitatively estimate the soliton-soliton offsets induced by an individual molecule adsorption event on the graphene surface. The graphene's Fermi level is given by $|E_F| = \hbar|v_F|(\pi N)^{1/2}$ with $\hbar = 6.582 \times 10^{-16}$ eV s. In our case, we have a carrier density $N = 2.94 \times 10^{16}$ m⁻² ($E_F = 0.2$ eV on silica) and an exposed area of 2×10^{-9} m². Let's now use single molecule NH₃ detection as an example. Each NH₃ molecule adsorption event will result in the transfer of two electrons from NH₃ to graphene, thus inducing a change in its carrier density and Fermi level $\Delta|E_F| = \hbar|v_F|(\pi)^{1/2}\{(N_1)^{1/2} - (N_2)^{1/2}\} \approx 4 \times 10^{-9}$ eV per molecule. If we now consider a wavelength ≈ 1650 nm, we obtain that the real part of graphene's refractive index n_g increases approximately by $0.8/\text{eV}$ [10.1364/OE.23.028170], and thus the n_g increment is $\approx 3 \times 10^{-9}$. Considering a typical beat note $f_A = 7.51$ MHz from the overlap between graphene and the mode pair TM₀₄ and TE₀₅, the adsorption of individual NH₃ molecules will induce an effective group index variation of $\approx 3 \times 10^{-13}$. Here the n_g dependent effective group index is calculated via a commercial software (*COMSOL Multiphysics*) using the finite element method (See supplementary note S2). Hence, the spectral variation of its soliton-soliton offset is ≈ 0.02 Hz per molecule (Eq. S7). Since the lock-in amplitude-frequency relationship is 10 mV/Hz, the adsorption of an individual NH₃ molecule will induce a step of ≈ 0.2 mV. This estimate is in excellent agreement with our experimental result.”

4. A minor comment with regard to the “theoretical maximum resolution” of 10^{-5} Hz: If not mistaken, this is based on a speculative 40 dB amplification of the 0.1 mV/Hz signal, without however considering the signal-to-noise ratio. I suggest to explain better why a useful 40 dB amplification is possible (considering the signal-to-noise ratio before and after amplification), or to remove the statement.

Response: We fully agree with the Referee that real maximum resolution is ultimately determined by the signal-to-noise ratio. As suggested by the Referee, we thus revise our claim in the new version of the manuscript:

“Finally, the lock-in amplification further enhances the sensitivity of our device.”

In addition, we have added the following explanation in section S8 of the supporting information:

We also quantitatively discuss the resolution limit of our scheme, when using a 7.5 MHz signal. As mentioned in the maintext, the detectable minimum frequency shift is determined by both the amplified ‘intensity-frequency’ relationship (unit: mV/Hz) and the noise of the amplified intensity (unit: mV). Using the amplification ratio of 40 dB, we measure the relative intensity noise (RIN) of the 50 kHz signal, as shown in **Fig. R2a**. Here the initial intensity is 1 mV. This clarifies that the RIN before and after amplification at low frequency (1 Hz level) is ≈ -50 dB. However, the RIN of the amplified signal at higher frequency is larger. In the measurement based on the lock-in amplifier and oscilloscope, such noise level induces an intensity uncertainty of $\approx 10^{-6}$. When the original signal sensitivity is 0.1 mV/Hz, the lock-in amplified intensity is 10 V, and the sensitivity can theoretically reach 1000 mV/Hz (or 10^{-3} Hz/mV), with uncertainty ≈ 0.12 mV. In **Fig. R2b** we show the temporal trace before and after amplification. Considering the 0.12 mV uncertainty, the limited resolution of this signal is 1.2×10^{-4} Hz, which is sufficient for individual molecule detection.

Fig. R2. Noise of the beat note signal. **a**, Relative intensity noise (RIN). **b**, Temporal trace of the twice down converted signal of the 7.55 MHz mode measured in an

oscilloscope. The blue curve represents the signal before amplification (1 mV) while the red curve shows the amplified signal (10 V).

Reviewer #3 (Remarks to the Author):

I commend the authors on the technical clarifications in the revised manuscript. One minor fact that the authors could consider is the correctness of Eq. S7 - it should use n_g , the group index instead of n_{eff} . With these revisions, the manuscript is, in my opinion, suitable for Nature Communications.

Response: We thank the Referee for recommending publication of the current version of the manuscript. In addition, we agree that the FSR is determined by the group index, typically denoted as n_g . In the previous version of our manuscript, we used the ' n_{eff} ' to avoid confusion with the graphene refractive index ' n_g '. In the revised version of the supplementary information we now use ' n_G ' instead of ' n_{eff} '.

REVIEWERS' COMMENTS

Reviewer #2 (Remarks to the Author):

The authors carefully addressed all comments and answered all questions in detail. This current version is a major improvement over the initially submitted version and I believe it can be published in this form.

Reviewer #3 (Remarks to the Author):

The authors have addressed the comments from reviewers in the previous round well. The addition of a figure demonstrating the repeatability and portability of the device strengthens the work substantially. I have no further comments, and recommend publication in Nature Communications.

Response Letter

All the editorial changes are marked with track-change in the revised version of the manuscript.

Reviewer #2 (Remarks to the Author):

The authors carefully addressed all comments and answered all questions in detail. This current version is a major improvement over the initially submitted version and I believe it can be published in this form.

Response: We thanks the Reviewer for the positive evaluation of the new version of our manuscript and for supporting publication of our work.

Reviewer #3 (Remarks to the Author):

The authors have addressed the comments from reviewers in the previous round well. The addition of a figure demonstrating the repeatability and portability of the device strengthens the work substantially. I have no further comments, and recommend publication in Nature Communications.

Response: We thank the Referee for recommending publication of our work and for acknowledging the technological impact of our proposed device.